# Bioarchitectural Design of Bioactive Biopolymers: Structure–Function Paradigm for Diabetic Wound Healing

**DOI:** 10.3390/biomimetics9050275

**Published:** 2024-05-04

**Authors:** Shivam Sharma, Anil Kishen

**Affiliations:** 1The Kishen Lab, Dental Research Institute, University of Toronto, Toronto, ON M5G 1G6, Canada; shivams.sharma@mail.utoronto.ca; 2Faculty of Dentistry, University of Toronto, 124 Edward Street, Toronto, ON M5G 1G6, Canada; 3Department of Dentistry, Mount Sinai Hospital, Toronto, ON M5G 1X5, Canada

**Keywords:** biopolymers, chronic wound healing, structure–function relationship, bioactivity

## Abstract

Chronic wounds such as diabetic ulcers are a major complication in diabetes caused by hyperglycemia, prolonged inflammation, high oxidative stress, and bacterial bioburden. Bioactive biopolymers have been found to have a biological response in wound tissue microenvironments and are used for developing advanced tissue engineering strategies to enhance wound healing. These biopolymers possess innate bioactivity and are biodegradable, with favourable mechanical properties. However, their bioactivity is highly dependent on their structural properties, which need to be carefully considered while developing wound healing strategies. Biopolymers such as alginate, chitosan, hyaluronic acid, and collagen have previously been used in wound healing solutions but the modulation of structural/physico-chemical properties for differential bioactivity have not been the prime focus. Factors such as molecular weight, degree of polymerization, amino acid sequences, and hierarchical structures can have a spectrum of immunomodulatory, anti-bacterial, and anti-oxidant properties that could determine the fate of the wound. The current narrative review addresses the structure–function relationship in bioactive biopolymers for promoting healing in chronic wounds with emphasis on diabetic ulcers. This review highlights the need for characterization of the biopolymers under research while designing biomaterials to maximize the inherent bioactive potency for better tissue regeneration outcomes, especially in the context of diabetic ulcers.

## 1. Introduction

“Form follows function” is a well-known principle coined by the architect Louis Sullivan [1] in the late 19th and early 20th century. This architecture and industrial design principle assert that the design of a building is tightly integrated to its intended purpose. This principle can be observed in nature as well, where organisms are designed to perform specific functions [2]. In the biological context, structure refers to the way components are arranged to create an organized system. Examples of biological structures include extracellular matrix, basement membrane, actin filaments, and DNA. Function, on the other hand, refers to the natural activities of an organism, which can exist at different levels within a biological system. The relationship between structure and function is bidirectional, meaning that the activity of an organism is determined by the organization of its components, while the organization of these components is also shaped by the activity they are meant to perform. This relationship can be expressed mathematically as follows:(1)Function=f(Structure)

The relationship between structure and function can exist at multiscales. A structure at a specific scale can have functions at much higher scales [3]. For instance, collagen crosslinking by advanced glycation end products (AGEs) [4] can result in altered functional consequences at hierarchical scales such as cell–matrix interactions [5], reduced modeling capacity in connective tissue and overall tendon function [6]. This structure–function paradigm has been widely described in various fields of biology such as protein engineering, tissue engineering, and evolutionary biology. Furthermore, this principle forms one of the foundational principles of designing organ-on-chip-engineering platforms for drug testing [7].

Natural bioactive materials or bioactives are natural substances that can cause a biological response [8]. The bioactives’ biological functionality is dictated by their structural properties, such as physico-chemical properties, surface chemistry, and conformation. There are three main categories of natural bioactives, which are based on their physical state. These categories include natural macromolecules or biopolymers, and secondary metabolites. Biopolymers and natural secondary metabolites can be produced by plants, animals, bacteria, or fungi. The biological potency of these substances is determined by the structural arrangement of their monomers to form polymeric chains and the chemical structure of the secondary metabolites [9,10,11,12]. The molecules in all categories possess the inherent potential for anti-oxidant, antibiotic, anti-diabetic, immunomodulatory, tissue reparative, and regenerative effects [13].

Chronic wounds are non-healing wounds that are characterized by prolonged inflammation, high proteases, and high reactive oxygen species (ROS) produced locally in the tissue. Diabetic ulcers are one such chronic wound that develops due to systemic hyperglycemia and neuropathy in diabetic patients [14,15]. Bioactives can have wide applications in treating chronic inflammatory conditions, especially diabetic wounds, due to their therapeutic bioactivity [16]. This article is a comprehensive review of the link between the inherent structural characteristics of natural bioactive biopolymers that are frequently used and their effectiveness in treating chronic wounds. It explains the structure–function relationship with a particular focus on the design tactics for wound healing materials based on biopolymers and not on bioactive secondary metabolites.

## 2. Natural Bioactive Macromolecules for Diabetic Wound Healing

Polymers are macromolecules composed of repeating units called monomers chemically bonded to each other primarily in a chain-like manner [17,18]. Polymers can be classified based on their origin into natural polymers, which are called biopolymers, and those chemically synthesized, called synthetic polymers [19]. Biopolymers are a class of naturally synthesized polymers derived from living organisms. They are composed of different monomeric units such as polynucleotides, which are made up of nucleotides like DNA and RNA, and polypeptides (proteins) that consist of amino acids and polysaccharides, which have carbohydrates as their monomeric unit [9]. These biopolymers are extensively used to design biomedical devices and biomaterials for tissue engineering solutions. They offer the advantages of biocompatibility, biomimicry, and biodegradability. Favourable mechanical properties allow them to fulfill the physical requirements of a scaffold, a delivery vehicle, or a substrate. Additionally, they possess intrinsic bioactivity that enhances the functionality of the bio-composites they are part of in terms of promoting skin regeneration. The commonly utilized natural biopolymers include alginate, chitosan, hyaluronic acid, collagen, gelatin, silk, fibrinogen, and cellulose [20]. These biopolymers mimic the micro-structural architecture of the natural ECM, facilitating cell–polymer interaction and allowing better cellular attachment and infiltration [19,20].

The intricate interplay between biopolymers and chronic wound tissue at the microstructural level orchestrates healing, driven by biochemical or biophysical cues that engage cellular responses and activate innate repair pathways [21]. Leveraging these cues, biopolymers with encoded biomimetic, instructive prompts establish an optimal regenerative microenvironment while forming essential barriers against moisture loss and infection, thus promoting favourable cellular responses [22]. These biopolymers exhibit bioadhesive, antimicrobial, anti-inflammatory, proangiogenic, and therapeutic delivery properties, serving as catalytic scaffolds for cellular interactions that foster proliferation and differentiation [23]. Additionally, these advanced biomaterials aid in exudate removal, provide protective cover, facilitate oxygen transport, and prevent infection [24].

Biopolymeric formulations have been developed where biopolymers act as excipients or active ingredients [25]. They exhibit versatility by transforming into many biomaterial platforms, such as hydrogels, hydrocolloid films, nanocomposites, electrospun nanofibers-based wound dressings, films, membranes, and nanoparticles, owing to their highly tunable physical and chemical behavior [26]. One notable advantage of these biopolymer formulations is their ability to absorb and hold water in a dry state, releasing water and other trapped molecules or drugs as a function of time, making them suitable for drug-delivery systems [27]. Immobilizing drugs or bioactives on these biomaterial platforms enhances drug specificity and efficacy while targeting tissues through their biological motifs [28]. Depending on their bioactivity, these biomaterials play an inherently pivotal role in tissue regeneration at different stages of wound healing. Moreover, their processing, purification, decellularizing, functionalizing and crosslinking techniques [13] will significantly influence these materials’ biological functionalities and responses. Biopolymers- alginate, chitosan, hyaluronic acid, and collagen, as the most popular biological macromolecules for skin repair and wound dressing applications have been well studied [29]. Extensive research was conducted to thoroughly relate the physical attributes of these four biopolymers to their wound healing bioactivity to finally achieve the desired wound dressing properties for chronic wounds.

Ideal properties to promote healing in chronic diabetic wounds should be considered for these biomaterials. Diabetic wounds, marked by their protracted healing, are frequently hindered by bacterial biofilms on the wound tissue. The presence of biofilm necessitates the antibacterial and anti-biofilm properties in these biomaterials. Additionally, chronic wounds exhibit elevated levels of oxidative stress and prolonged inflammation. Hence, it is crucial to have free radical scavenging, anti-oxidation, and anti-inflammatory properties by modulating the inflammatory response. An inherently anti-diabetic substance would offer benefits in regulating glucose levels, directly targeting the hyperglycemic state of these wounds. Core to wound healing is angiogenesis and cellular migration, which are processes impaired in diabetic wounds. Therefore, an ideal biomaterial should promote the formation of new blood vessels to increase the blood flow and oxygenation at the wound site. The ideal biomaterial should combine these properties to promote the healing of diabetic wounds [30]. Achieving these properties requires engineering the physico-chemical characteristics at the chemical and molecular scale and aligning them with the desired bioactivity for diabetic wound healing (Figure 1). This requirement underscores the significance of understanding the structure–function paradigm in biopolymers.

### 2.1. Structure–Function Paradigm in Alginate

Alginate is an anionic polysaccharide obtained from brown algae’s (*Phaeophyceae*) cell walls (*Laminaria* sp., *Ascophyllum* sp., and *Macrocystis* sp.) [31,32,33]. As a component of the cell wall, it provides the flexibility and mechanical strength necessary to withstand the natural forces of water in its natural habitat [34]. It is a linear co-polymer, having (1,4)-linked D-mannuronic acid and L-guluronic as its monomers in different combinations [33], giving rise to diverse structures, physico-chemical properties, and molecular weights. The source and extraction methods influence the average molecular weight (MW) of alginate biopolymer, falling within the range of 32–400 kDa [35]. The pKa of alginate significantly diverges from that of its individual monomers, D-mannuronic acid (pKa 3.38) and L-guluronic (pKa 3.65), in 0.1N NaCl. The polymer concentration and the solvent’s ionic strength determine the pKa of alginate [36]. The mannuronic acid residue has ^4^C1 conformation compared to guluronic acid, which has ^1^C4 conformation in the backbone of alginate, irrespective of the monomeric ratios [37]. Alginate degrades into linear low MW polymeric chains (2–25 monosaccharides) called alginate oligosaccharides. Although alginate is inherently non-degradable in mammals as they lack the enzyme (i.e., alginase) capable of cleaving the polymer chains, its minimal immunogenicity and non-toxic properties provide an advantageous quality for its utilization in biomedical research. Based on monomer sequences, these oligomers can be three types: oligo-mannuronic acid (Oligo-M), oligo-guluronic acid (Oligo-G) and hetero-oligomeric mannuronic, and guluronic acid (Oligo-MG) [38]. Structure-related parameters such as monomer sequences, MW, degree of polymerization (DP) and end-group structure and modification affect their biological activities that help in directional synthesis according to the desired application [38]. The role of these oligosaccharides in different structural forms for their antitumor, anti-diabetic [39,40], antimicrobial [41] anti-inflammatory [42], and anti-oxidant [43] properties have been explored.

Oligo-M forms a stretching chain-like structure due to equatorial conformation of β-1,4-glycosidic bonds compared to Oligo-G, which forms a helical structure due to axial conformation of α-1,4-glycosidic bonds [44]. Monomer sequences can precisely alter the antibacterial activity of oligosaccharides. Oligo-M (4.24 kDa) has direct antibacterial effects as shown by inhibiting the growth of pathogenic bacterial species such as *E. coli* and *S. aureus* [45]. Oligo-G’s antibacterial effect is that it destroys the formation of biofilms [41]. For instance, Oligo-MG with a higher G ratio exhibited significant anti-biofilm effects. Oligo-G CF5/20 having G (85%)/M (15%) perturbed the *P. aeruginosa* biofilm formation [41,46,47] and Oligo-G (90–95%) (2.6 kDa) could potentiate the activity of antibiotics by destroying biofilm [48]. Another mechanism in which Oligo-G exhibited its antibacterial properties was by promoting microbial aggregation and hindering bacterial motility through its interaction with the bacterial surface, resulting in the disruption of bacterial integrity [48,49]. Oligo-G has more substantial antibacterial effects than Oligo-M (3–3.5 kDa) and Oligo-MG (4–4.5 kDa), indicating its potential use as a carrier for various antibiotics [50]. However, higher oligo-M content showed stronger immune regulation [51] and anti-oxidant properties [52]. For example, alginate with a higher M/G ratio induced higher TNF-α secretion from RAW264.7 cells [53].

The structural conformation of Oligo-G that binds to a specific pattern recognition receptor (PRR)–Toll-like receptor (TLR) 4 promotes cytoskeletal remodeling and macrophage proliferation via activation of downstream nuclear factor (NF)-κB/Akt/mammalian target of rapamycin (mTOR)/mitogen-activated protein kinase (MAPK) signaling pathways, resulting in an improved innate immunity response [54]. Furthermore, the Chromium (III) complex formed with Oligo-M (2.8–3.2 kDa) boosts glucose uptake by activating both the insulin signaling pathway and the adenosine monophosphate-activated protein kinase (AMPK) pathway, thereby enhancing insulin sensitivity [40]. Oligo-MG inhibits oxidative stress and apoptosis in human umbilical vein endothelial cells (HUVECs) [55] via activation of the integrin-α/focal adhesion kinase (FAK)/phosphoinositide 3-kinases (PI3K) pathway and regulation of the expression of phosphatase and tensin homologue chromosome 10 (PTEN), P21, and cyclin-dependent kinase (CDK2) [43]. Anti-oxidant and anti-inflammatory effects of Oligo-MG were observed as an increase in serum levels of Superoxide Dismutase (SOD) and Glutathione (GSH) and a decrease in interleukin (IL)-1β [56].

Structural modification of the alginate oligosaccharides with active ions such as sodium, Vanadium, and Chromium [44] can improve their biological activities. For instance, modification by Vanadyl leads to an increase in anti-oxidant activity [57], the Chromium (III) complex of Oligo-M increases insulin sensitivity as described above [40], and Seleno-modification improves anti-oxidation and anti-inflammatory effects [42,44]. Seleno-polymannuronate significantly attenuated the nitric oxide production, prostaglandin E_2_ (PGE_2_), reactive oxygen species (ROS), inducible nitric oxide synthase (iNOS) and cyclooxygenase-2 (COX-2) expression, and pro-inflammatory cytokines via supressing NF-κB and MAPK signaling pathways in lipopolysaccharide (LPS)-induced RAW 264.7 macrophages. These anti-inflammatory effects were also observed in primary murine macrophages stimulated with LPS [42]. On the contrary, alginate stimulates macrophages to produce pro-inflammatory cytokines, such as IL-1β, IL-6, IL-12, and tumor necrosis factor-α (TNF-α) via NF-κB activation similar to pathogen recognition and LPS stimulation in a dose and time dependent manner, as seen in in vitro studies on RAW 264.7 macrophages [58].

DP and MW of the oligosaccharides can affect the bioactivity of the biopolymer. Low MW alginate oligosaccharides exhibit good anti-coagulant properties by inhibiting thrombus formation [59] and also act as anti-oxidants. Oligosaccharides with DP 2–5 induce higher nuclear translocation of NF-κB p65 subunit and higher nitric oxide (NO) production [60]. Oligo-M with DP 7 and Oligo-G with DP 8 have the most suitable molecular size and structural conformation to act as ligands of receptors on macrophages for IL-1α, IL-1β, IL-6 and TNF-α secretion. Additionally, the presence of TLR 2 and TLR 4 antibodies have the potential to suppress the activity of these Oligo-M (DP 7) and Oligo-G (DP 8), indicating that these oligosaccharides may trigger PRRs on macrophages [61].

Enzymatic alginate degradation generates oligosaccharides with unsaturated terminals, whereas chemical degradation generates oligosaccharides with saturated terminals [44]. Unsaturated oligosaccharides are better anti-oxidants than saturated ones primarily because additional functional groups have an alkene acid structure [62,63]. Unsaturated Oligo-G and Oligo-M induce NO and iNOS expression, cause ROS and TNF-α production via NF-κB/MAPK pathways activation and induce macrophage phagocytic bactericidal activity [64] compared to saturated oligos. Unsaturated Oligo-M are better inducers of cytokines- TNF-α, granulocyte-colony stimulating factor (G-CSF), monocyte chemoattractant protein-1 (MCP-1) in comparison to unsaturated Oligo-G [65]. Saturated oligosaccharides alleviate the inflammatory response in macrophages stimulated by LPS via decreasing the LPS binding to the cell surface, thus eliciting anti-inflammatory effects. Ultimately, it attenuates NO, PGE_2_, ROS, iNOS, and COX and pro-inflammatory cytokines- TNF-α, IL-1β, and IL-6 [66].

Alginates are biocompatible, have low toxicity [67], and have the property to be quickly crosslinked (by divalent cations, e.g., calcium) [68] and/or conjugated with other small or macro-molecules, making them a good candidate for wound dressings and drug-delivery vehicles [69]. The formation of hydrogen bonds between water molecules and polar groups (OH and COO-) on the alginate chain makes it hydrophilic. This enables it to retain the body fluids, contributing to maintaining a moist wound environment [70]. MW and block composition of alginate affect its biological response via altering its polyelectrolyte nature. Physically crosslinked electrospun alginate nanofibrous membranes showed cellular adhesion with fibroblasts, keratinocytes, and osteoblasts underscoring their potential applications in skin and bone regeneration [71]. Cellular adhesion has been further improved by functionalizing alginates with cell adhesion motifs [72] or crosslinking them with ECM proteins [73]. Similarly, in vivo studies have shown that calcium alginate scaffolds facilitate platelet and erythrocyte aggregation [67,74]. Alginate has been combined with other natural biopolymers such as gelatin [75], hyaluronic acid [76,77], and chitosan [78] to create composites tailored to promote healing. In a recent study, researchers formulated a blend of physically crosslinked sodium alginate and carboxymethyl cellulose film, aiming to enhance water retention, slow degradation, and improve mechanical strength. These attributes promoted efficient gaseous exchange between the diabetic wound bed and the surroundings, leading to increased collagen deposition, re-epithelialization, and accelerated wound closure in a diabetic rat model [79]. The structural similarity of alginate to wound tissue ECM makes them an attractive wound healing material, especially for treating and managing diabetic wounds [80,81].

Due to its suitability as a delivery vehicle and inherent bioactivity of alginate, recent studies have shown alginate-based biomaterials to be quite effective in healing diabetic wounds. For instance, a pre-clinical study showed that calcium alginate gel alone and loaded with a microelement (Rubidium) could synergistically enhance angiogenesis, re-epithelialization, and collagen deposition in diabetic wounds [82]. In support of that, owing to its angiogenic and matrix deposition potential, a composite alginate hydrogel was designed to enhance the wound healing rate [83]. ECM deposition by calcium alginate is due to the exchange of calcium ions from the polymer with sodium ions in the wound exudate/blood inducing the production of platelet-derived growth factor (PDGF) and cell-recruiting cytokines [84]. In an animal skin wound healing model, the use of topical alginate treatment has demonstrated enhanced healing outcomes. This improvement linked to the expression of transforming growth factor-β1 (TGF-β1), fibronectin, vascular endothelial growth factor (VEGF), and collagen-I [85]. To further accelerate the wound healing process, a dual network alginate containing plasma-rich protein (PRP) hydrogel was designed to deliver epidermal growth factor (EGF) and VEGF [86].

In summary, the diverse bioarchitecture and associated bioactive properties of alginate emphasize its potential as a multifunctional biomaterial for chronic wound healing (Figure 2). The intricate interplay between alginate’s molecular characteristics, such as monomer sequences, MW, and DP, influences its biological activities, including antibacterial, anti-inflammatory, anti-oxidant, and wound healing effects. Furthermore, modifications with active ions and functional groups offer avenues for enhancing alginate’s bioactivity and targeted therapeutic interventions. Alginate-based biomaterials exhibit excellent biocompatibility and the ability to interact with biological systems, making them versatile candidates for wound dressings, drug-delivery systems, and tissue engineering scaffolds.

### 2.2. Structure–Function Paradigm in Chitosan

Chitosan is one of the most abundant natural polysaccharides obtained from chitin present in the exoskeleton of crustaceans, molluscs, and fungal cell walls. In its natural form, chitin functions as a vital structural element exclusively found in ECM, such as the cell walls of fungi, mollusk shells, and arthropod cuticles and offers both protection and structural support to the organism, contributing to its overall integrity and resilience [87]. Chemically, it is formed by deacetylation of chitin to form a linear chain of randomly distributed β-linked D-glucosamine and N-acetyl-D-glucosamine [88]. Most commercially available chitosan typically have MW in the range of 50–2000 kDa and exhibit an average degree of deacetylation (DDA) between 50 and 100%, commonly 80–90% [89]. It is soluble in acidic aqueous solution through primary amine protonation having pKa 6.5, whereas chitin is insoluble in water due to high number of acetylated residues [90]. The biological activity of chitosan is tightly associated with its physico-chemical properties such as MW, degree of acetylation (DA), DP and functional groups on the polymer backbone [91].

Chitosan is well studied and used for its antimicrobial activity that is directly related to the DA, MW, cationic charge density, degree of substitution, and hydrophobicity. The antimicrobial functionality is due to the electrostatic interaction of positively charged -NH_2_ groups from the polymeric chain with the negatively charged COO- groups of the bacterial cell wall or outer membrane. This can lead to leakage of the intracellular components in the case of gram-positive bacteria and reduce permeability of the outer envelope, blocking the nutrient access for gram-negative bacteria [92]. Chitosan could also intracellularly block the RNA transcription after binding with bacterial DNA [93]. Chitosan could be chemically modified into various derivatives to have enhanced biocidal activity. One such derivative–quaternised chitosan has been reported to have enhanced antibacterial activity against *S. aureus*, *E. coli*, and *C. albicans* [94]. Its biocompatibility and antibacterial potency could be controlled by adjusting the quaternary ammonium’s substitution degree. A study showed that hydroxypropyltrimethyl ammonium chloride chitosan (HACC) with 18% and 44% quaternary ammonium had higher antibacterial activity against *S. aureus*, Methicillin-resistant *Staphylococcus aureus* (MRSA), and *S. epidermidis* compared to 6%. However, HACC with 6% and 18% substitution was non-cytotoxic towards mouse fibroblasts (L929) and did not hinder proliferation and osteogenic differentiation of human bone marrow stromal cells (hMSCs) [95]. Quaternized chitosan shows better antifungal effects than pristine chitosan. A structure–function paradigm in quaternised chitosan was exhibited where MW could influence the antifungal properties, and, as MW increases, more substantial antifungal properties are observed [96].

The diverse array of functional groups along the chitosan polymer chain allows its conjugation with other bioactive compounds or drugs, enhancing their efficacy for various biomedical applications. Primary amino groups and primary and secondary hydroxyl groups serve as the principal reactive sites on chitosan for such interactions. These groups allow chitosan modifications via metal coordination, chemical coupling, crosslinking, co-polymerization, sulfonylation, and alkanoylation giving rise to new properties [90]. Chitosan, along with its derivatives [97,98] and oligosaccharide forms [99], also act as excellent anti-oxidants due to these functional groups that can effectively scavenge free radicals. Chito-oligosaccharides could inhibit ROS production and oxidative stress-induced apoptotic pathways, including inhibiting NF-κB and MAPK pathways, and regulate the Bcl-2-associated X protein (Bax)/B-cell lymphoma protein 2 (Bcl-2)/Caspase-3 signaling pathway and activation of the nuclear factor erythroid 2-related factor 2 (Nrf-2)/Kelch ECH associating protein 1 (Keap1) signaling pathway [100]. Chitosan has natural anti-inflammatory properties that are also contingent on the structural parameters like MW and DA. LPS-induced RAW 264.7 macrophages when treated with chitosan with higher MW (300, 156, and 72 kDa) showed a significant reduction in NO, TNF-α, and IL-6 as compared to lower MW (7.1 and 33 kDa) and oligosaccharides (DP 1–5). These differences in anti-inflammatory properties as a function of MW of chitosan were due to variations in its binding receptor and the affected downstream molecular pathway [101]. Moreover, changing the DA could influence the chemotactic behavior of neutrophils as they were dose-dependently attracted to 80% deacetylated chitosan, whereas 95% deacetylated did not show any chemotactic effects [102]. DP of chito-oligosaccharides also affects the inflammatory response in biological systems. For example, chito-oligosaccharides with DP 1–6 have shown to produce more NO in Interferon-γ (IFN-γ)-induced RAW 264.7 macrophages [103], whereas, in another study, NO levels were reduced by chito-oligosaccharides having DP 2–6 in LPS-induced RAW 264.7 macrophages [104]. However, chito-oligosaccharides with DP > 4 had an inhibitory effect on cytokines- IL-1β, IL-17A, and IFN-γ levels produced by LPS-stimulated RAW 264.7 macrophages [105].

Chitosan and its derivatives, including oligosaccharides, have hypoglycemic and anti-diabetic effects. They can hinder adipogenesis (transforming pre-adipocytes to adipocytes, hence, suppressing fat accumulation), inhibit intestinal carbohydrate-hydrolysing enzyme (thereby, reducing the glucose absorption of glucose in the blood), protect pancreatic β-cells, and enhance insulin secretion [106]. Hyperglycemia in diabetes could lead to glucose toxicity and eventually cause oxidative stress through the generation of free radicals in the body that are detrimental to insulin-secreting β-cells in the pancreas [106]. Various pre-clinical studies have shown that chitosan and its derivatives can protect the pancreatic β-cells against free radicals by boosting the anti-oxidant enzyme production or by scavenging them [107]. Mechanistically, chitosan’s hyperglycemic actions against diabetes mellitus are induced by PI3K/Akt signaling pathway that controls pancreatic β-cell activity, insulin secretion, and glucose metabolism [108]. It promotes glucose uptake by augmenting Glucose Transporter 2 (GLUT 2) [109] and Glucose Transporter 4 (GLUT 4) [110].

Chitosan is antibacterial, anti-inflammatory, and anti-diabetic, as discussed above, and it is also biodegradable, biocompatible, hemostatic, and mucoadhesive, all owing to its structural properties [111]. Increasing the degree of quaternization (21.1–48.8%) in N-trimethyl chitosan chloride (TMC), leading to conformational changes and a decrease in polymer-chain flexibility, had detrimental effects on mucoadhesion [112]. Electrostatic attraction between the cationic chitosan chain and anionic mucin emerges as the primary mechanism responsible for chitosan mucoadhesion, complemented by additional contributions from hydrogen bonding and hydrophobic interactions [113]. These functionalities make this biopolymer to be very promising for diabetic wound healing.

Chitosan dose, MW, DA, and acetylation clustering patterns (block and random) can be tuned to modulate the macrophage immune response. Distinct chitosan formulations, such as 10 or 190 kDa with 80% block DDA and 3, 5, or 10 kDa with 98% DDA, exhibited the ability to stimulate macrophages to release CXCL10 and Interleukin-1 Receptor Antagonist (IL-1RA) at 5–50 μg/mL via IFNβ paracrine activity and STAT 2 activation. On the contrary, at a dose of 50–150 μg/mL, they activated the inflammasome signaling, leading to the release of IL-1β and PGE_2_. Structural motif in chitosan that is required for a cytokine response is ≥3 kDa block glucosamine, and lysosome rupture is the key mechanism to determine the IL-1β or IL-1RA secretion. Chitosan with 80% and 95% DDA having different MW has been documented in distinct studies as capable of inducing both pro- [114,115,116,117,118] and anti-inflammatory [116,119] responses in macrophages. More recently, studies have revealed strong immunomodulatory and intercellular crosstalk regulatory effects of chitosan [120]. Chitosan nanoparticles have been shown to reduce pro-inflammatory molecules (NO and IL-1β) in THP-1 monocytes-differentiated macrophages cultured alone and co-cultured with fibroblasts. It was also demonstrated that these nanoparticles could facilitate macrophage polarization towards an M2-like phenotype via modulating the crosstalk between these two cells [121]. The immunomodulation was further confirmed by macrophage proteomic profiling, which revealed the reduction of pro-inflammatory markers (NO and IL-1β), the enhancement of anti-inflammatory cytokines (IL-10 and TGF-β1), and the upregulation of immunoregulatory and anti-oxidant proteins [122]. The discrepancies in immune response could be attributed to partial structural characterization and variation in the dosage of chitosan, making it essential to characterize the structural properties while establishing its immunomodulatory effects.

Various chitosan-based biomaterials have been developed for wound healing in the form of hydrogels, membranes, nanoparticles, films, and sponges [123,124]. During early wound healing events, chitosan show exceptional hemostatic characteristics [125,126]. This non-specific binding can be attributed to the positively charged amino group on the polymer chain and plasma membrane [127,128]. Chitosan-based dressings developed by Medline Industries (Chicago) effectively controlled bleeding in rat wounds [129]. Chitosan dressings on third-degree burns in mice could promote the formation of granulation tissue while promoting higher TGF-β1 expression in early stages [130]. Chitosan scaffolds with and without basic Fibroblasts Growth Factor (bFGF), which were used to treat chronic pressure ulcers in aged mice, supporting its neutrophil recruiting activity. It was demonstrated that both scaffolds significantly accelerated healing by elevating neutrophil levels and inhibiting neutrophil elastase activity [131].

In terms of its application in chronic wounds, in vitro and in vivo investigation on chitosan–genipin hydrogel is antibacterial and effective in healing pressure ulcers by eliciting an increased immune response, thereby enhancing the migration and proliferation of keratinocytes and fibroblasts [132]. In a diabetic mice model, it was shown that collagen/chitosan-based composite gels increased collagen deposition, hair follicle regeneration, and sebaceous gland formation. It has been highlighted that the enhanced growth factors and cytokines such as VEGF, TGF-β1, IL-1β and Tissue Inhibitors of Metalloproteinases 1 (TIMP1) gene expression are the reason behind the mechanism [133]. In another study, a composite nanohydrogel consisting of ferulic acid-grafted chitosan (CS-FA), oxidized hydroxyethyl cellulose (OHEC), and green synthesized selenium nanoparticles, showed promising anti-oxidant, antibacterial, and wound healing properties both in vitro and in a diabetic wound animal model [134]. Furthermore, experiments conducted on diabetic leg ulcers’ exudates have shown that the gallic acid crosslinked–chitosan hydrogel has the ability to stop Matrix Metalloproteinases (MMPs) by binding zinc ions to their active center and also to capture free radicals [135].

Derived from chitin, chitosan’s bioarchitectural variations, including MW, DP, and DDA, play pivotal roles in determining its interactions with biological systems and therapeutic efficacy (Figure 3). From antimicrobial and immunomodulatory to anti-diabetic properties, chitosan showcases a multifaceted nature that holds promise for addressing various complex challenges of chronic wounds in diabetic patients. By harnessing the structure–activity relationships of chitosan, researchers and clinicians can develop innovative wound healing therapies that improve patient outcomes and enhance their quality of life.

### 2.3. Structure–Function Paradigm in Hyaluronic Acid

Hyaluronic acid (HA) or hyaluronan is an anionic non-sulfated glycosaminoglycan (GAG) distributed extensively throughout neural tissues, synovial fluids and epithelial and connective tissues [136]. It was first isolated from the vitreous body of the bovine eye [137]. It is found in many strains of bacteria and is omnipresent in vertebrates, located explicitly in embryonic tissues and the ECM of connective tissues [136]. Chemically, it is a highly hydrophilic biopolymer composed of disaccharides, β-1,4-glucuronic acid, and β-1,3-N-acetyl glucosamine units linked via alternating β-(1→4) and β-(1→3) glycosidic bonds [138]. Its MW varies depending on the source, but it can reach up to 2 × 10^7^ Da. Carboxylic groups in the polymeric chain of HA make it anionic and, hence, highly hydrophilic. High MW and hydrophilicity allow it to form a viscous network essential to ECM regulating tissue homeostasis and resistance to compressive forces. Due to its hydrophilic nature, it possesses a significant capacity to retain water and exhibit lubrication properties. As the MW, concentration and shear rate increase, and HA in an aqueous solution transitions from exhibiting Newtonian to non-Newtonian fluid characteristics. Higher MW and concertation correspond to increased viscoelasticity [139]. The viscoelastic properties of HA are influenced by pH and the ionic strength of the solvent. With a pKa of 3, alterations in pH impact the degree of ionization and intermolecular interactions, consequently modifying its rheological behavior [140].

Structural properties of HA such as MW, degree of substitution, and crosslinking density can have a significant impact on its mechanical and rheological properties, diffusion profile, and permeation rate, due to which the wide range of applications in biomedicine, from bio-revitalizing skin cosmetics [141] to wound dressings, can be explored [142,143]. The bioadhesive and viscoelastic features of HA, dependent on its MW, can significantly impact drug entrapment, release, and skin penetration in HA-based topical drug-delivery systems [144,145,146]. In instances with high MW (>17,000 kDa), HA reduction in drug mobility and penetration was observed due to molecular chain entanglement compared to lower MW HA when the mentioned solutions were used [147]. A comparative analysis of HA’s mucoadhesive and penetration-enhancing properties with different MWs (202, 693, and 1878 kDa) revealed that lower MW HA demonstrated superior mucoadhesive efficacy [148].

HA also has differential bacteriostatic and bactericidal effects based on its MW and concentration [149]. High MW HA better minimizes bacterial contamination when tested on microbes in the planktonic phase. HA interferes with ligand receptor interaction involved in bacterial attachment, reducing bacterial adhesion and biofilm formation [150]. The high MW HA increases bacteriostatic activity by saturating bacterial hyaluronidase with an abundance of HA [151]. This prevents bacterial proliferation due to the significant breaking down of ECM components [152].

HA interacts with proteoglycans to form proteoglycan aggregates that crosslink with other matrix proteins, such as collagen, forming supramolecular structures and the stabilizing gel state of the ECM [153]. At the cellular level, HA forms the pericellular coat acting as a signaling molecule to regulate cell adhesion, migration, and proliferation [136]. HA binds to specific cell surface proteins called hyaladherins [154,155]. Two primary hyaladherins are CD44 and the Receptor for Hyaluronan-mediated Motility (RHAMM) or CD168, through which intercellular signaling pathways associated with proliferation, differentiation, and cell motility are activated [156]. The HA-CD44 affinity is concentration and MW-dependent on glycosylation and phosphorylation of the serine residues on protein domains [157]. High MW HA can cluster the CD44 receptor on the cell membrane, allowing the receptor to interact with other ligands such as ECM proteins, MMPs, cytokines, and growth factors modulating the receptor activity [158,159]. HA-CD168 interaction mediates cell migration via interaction with skeletal proteins during tissue repair and inflammation processes [160].

Structural properties, particularly MW and the polymeric chains’ functional group, directly influence the biological functionality [161,162]. A structure–function relationship exists, as the MW dictates the HA’s potential of immunomodulation and immunoprotection. High MW HA, more than 1000 kDa, has filling and hydrating functions as a component of soft connective tissue. Moreover, due to its electrostatic repulsion by the negatively charged carboxyl acid groups, it creates a swollen hydrated network crucial to embryonic development and tissue organization. High MW HA (>800 kDa) promotes an alternatively activated-like state by up-regulating pro-resolving gene transcription, including *arg1*, *il10*, and *mrc1*, and enhanced arginase activity [163]. Therefore, it has an anti-inflammatory effect and promotes epithelial cell homeostasis and survival. An amount of 1 mg/mL dose of high MW HA inhibits inflammatory cell chemotaxis, phagocytosis, elastase release, and respiratory burst activity [164]. In addition to being anti-inflammatory, it also acts as an anti-fibrotic agent in rheumatoid and osteoarthritis [165] and tympanic membrane perforation healing [166]. In addition, exogenous administration of HA has also been shown to enhance cutaneous healing through a feedback loop, promoting cell proliferation and migration, and tissue hydration [167]. High MW HA in the pericellular matrix of pancreatic islets protects insulin-secreting pancreatic β-cells from leukocyte-mediated death. High MW HA binds to CD44 receptor on regulatory T (T_reg_) cells and increases the expression of transcription factor Foxp3 and induction of anti-inflammatory cytokine, IL-10, which ultimately leads to increased immune suppression capacity [168]. Furthermore, high MW HA hydrogels demonstrated reduced leukocyte infiltration and enhanced blood vessel formation in oral wound healing following third molar extraction [169]. On the contrary, low MW HA or fragmented HA are pro-inflammatory by triggering TLR-activated inflammatory signaling cascade [170].

In its pristine form, HA exists as a high MW polymer. However, during inflammation, low MW fragments tend to accumulate. Inflammatory genes such as macrophage inflammatory protein-1 α (MIP-1α), macrophage inflammatory protein-1 β (MIP-1 β), cytokine responsive gene-2, and monocyte chemoattractant protein-1 (MCP-1) were activated in macrophages by these fragments (hexamer, 35 kDa, 280 kDa) [171,172]. Furthermore, low MW HA (<5 kDa) induces a classically activated-like state as evidenced by up-regulation of pro-inflammatory genes, such as *nos2*, *tnf*, *il12b*, and *cd80*, and increased production of NO and TNF-α [163]. It was further supported by HA induction of macrophages via activation of CD44 and leading to TNF-α-mediated secretion of insulin-like growth factor-1 (IGF-1) and IL-1β [173]. Unlike native high MW HA, which is reported as an anti-angiogenic, low MW HA (~250 kDa) and oligomers with 6–10 units are highly pro-angiogenic [163]. However, 4-unit HA fragments could not prompt an angiogenic response [174]. When HA macromolecules interact with CD44 and CD168 receptors on endothelial cells, it triggers collagen synthesis and the steps of angiogenesis, i.e., proliferation, migration, and cell sprouting [175]. Another proven mechanism could be by induction of plasminogen activator-inhibitor-1 (PAI-1) via RHAMM-TGF-β Receptor 1 (TGFβR1) signaling in endothelial cells [176]. While low MW HA can permeate through the skin, topical administration showed that it could prevent granulation tissue from oxidative damage during wound healing process [177,178]. Intermediate or medium MW HA (100–300 kDa) promotes wound healing in the proliferative phase of healing process. It has shown to increase the epithelial tight junction protein Zonula occludens (ZO-1) and Purinoreceptor (P2X7) basal activation in keratinocytes for improved intercellular interaction and proliferation leading to faster wound closure [179].

Among diabetic individuals, elevated plasma levels of HA are observed due to hyperglycemia. Platelets break down this plasma HA into lower MW fragments, partially associated with impaired healing [180]. The low MW fragmented HA can lead to leukocyte homing and adhesion, initiating inflammatory and angiogenic signaling while inducing IL-6 and IL-8 [180]. These low MW HAs could be used to heal diabetic wounds. A topical ointment containing oligomers of HA (2–10 disaccharide units) significantly increased endothelial cell proliferation, migration, and tube formation under high glucose conditions. The mechanism that led to increased angiogenesis in the wounded skin explained is increased Src/extra-cellular signal regulated kinase (ERK) transduction signaling and TGF-β1 expression [181].

As detailed above, the structure-dependent immunomodulatory and antibacterial properties influence cellular adhesion, proliferation, and migration [161]. Hence, it can have temporal effects in the inflammation and proliferation (granulation and re-epithelialization) stages of chronic wound healing [182]. During the early phases of routine wound healing, HA is produced in large quantities as a component of ECM that interacts with various growth factors and cytokines required to facilitate cell migration. Additionally, it acts as a substrate for adhesion, migration and proliferation of fibroblasts, keratinocytes and endothelial cells [183,184], and increases angiogenesis to promote scar-less healing [185]. Highly sulphated hyaluronan nanoparticles loaded in a collagen/HA hydrogel downregulated pro-inflammatory macrophage activities in vitro, ex vivo, and in vivo, increased vascularization, and accelerated new tissue formation in a mice model of acute skin inflammation [186]. Recently, 3D bioprinting methods have been employed to create HA–collagen composites (MW 100–200 kDa) incorporated with silver nanoclusters for addressing chronic diabetic wounds. These hydrogels have demonstrated the ability to stimulate fibroblast proliferation and migration, while also augmenting collagen deposition, as evidenced by both in vitro and in vivo investigations [187]. MW dependency is also reported on wound healing outcomes. For instance, the role of HA formulation in combination with a povidone–iodine complex was explored where it was demonstrated that higher MW HA (1000 kDa) significantly reduced wound area, increased neo-vessels and epithelialization via increasing VEGF production and suppressed inflammatory response in diabetic rats [188]. In vivo evaluation of a mouldable and self-healing supramolecular HA hydrogel on a rat full-thickness skin wound model revealed a high epidermal regeneration rate [189].

HA stands out as a multifaceted biopolymer with distinctive structural properties, particularly its MW and functional groups that dictate its role in modulating inflammatory responses, promoting tissue regeneration, and enhancing angiogenesis (Figure 4). Moreover, HA exhibits differential effects on inflammation and immune response depending on its MW that could be beneficial for impaired healing conditions like diabetic wounds. Through its interactions with cell surface receptors and growth factors, HA facilitates cellular adhesion, migration, and proliferation, crucial processes in the granulation and re-epithelialization phases of wound healing. HA-based formulations, ranging from nanoparticles to hydrogels, offer tailored approaches for promoting wound closure, reducing inflammation, and accelerating tissue repair. As research continues to unveil the intricate mechanisms underlying HA’s therapeutic efficacy, it holds immense promise for revolutionizing diabetic wound care and advancing strategies for managing chronic wounds effectively.

### 2.4. Structure–Function Paradigm in Collagen

Collagen is the most abundant protein in the human body and is the main component of ECM. Its tertiary structure comprises three polypeptide chains (mainly composed of amino acids like proline, hydroxyproline, and glycine) wrapped in a left-handed triple helix to form collagen fibrils [190]. The process of collagen fibrillogenesis is strongly driven by increased entropy, resulting in a greater level of disorder at the water–protein interface [191]. Fibroblasts synthesize collagen de novo an integral part of normal wound healing process [192]. Although collagen fibrillogenesis can be induced in vitro, the resulting collagen networks typically consist of randomly oriented fibers, unlike the highly aligned bundles found in tissues with exceptional tensile strength, such as bones and tendons [193]. These aligned bundles enable collagen to resist stretching and deformation. Helix formation during fibrillogenesis is reliant on electrostatic interactions along the collagen fibrils. Thus, it becomes susceptible to environmental factors that can alter the charge distribution along the collagen molecule. Factors such as pH and ionic strength of the buffer, as well as the inherent structure of the collagen molecule, play crucial roles in this process. When pH significantly deviates from the isoelectric point of collagen molecules, fibrillogenesis tends to be slower or partial, leading to the loss of characteristic D-banding [194,195].

Additionally, the amino acid sequence of the collagen molecule exerts various influences on fibrillogenesis. The packing of collagen fibrils depends on changes in amino acid conformation along the chain [196]. This structural variability translates into differences in assembly rates between collagen types. For instance, collagen type III and II form fibrils more rapidly than type I due to the higher molecular mass of type I and fewer intermolecular interactions [193]. The source of collagen also introduces variability in amino acid structure and inherent biological signaling. Notably, the most variable regions within mammalian collagen chains are the peptide sequences proximal to protein-binding sites, which can modulate specific cell responses [197]. The structural characteristics of collagen are of paramount importance since they dictate the availability of signaling cues, such as adhesion ligands, for cellular communication. Consequently, modifications to collagen structure can be strategically employed to direct cellular responses. Its hydrophilic nature makes it possible to blend with other natural (dextran [198], chitosan [199], alginate [200], HA [187]), and synthetic polymers (polyacrylamide [201], polyvinyl alcohol [202]) to make stable formulations. Collagen, considered to be a ‘self’ molecule, is immunologically accepted by the human body and is also highly biocompatible and biodegradable by endogenous collagenases. Hence, it can be utilized for various wound dressings (hydrogels, electrospun fibers, or nanocrystal-containing scaffolds) [203].

The functional role of collagen exhibits variability depending on its structural attributes [204,205,206]. These structural attributes in context of collagen encompass chemical alterations in its structure (including crosslinking, glycation, and non-covalent interactions), the types of suprastructures formed, and triple-helix content. The extent of collagen crosslinking increases its tensile strength and resistance against proteolytic degradation. Hence, this phenomenon is leveraged by utilizing external agents such as chitosan nanoparticles to crosslink the collagen and stabilize it against bacterial proteolytic damage [207]. Glycation of collagen fibrils by glucose represents another chemical modification in the aging process and diabetes. This process is a spontaneous, non-enzymatic reaction between the carbonyl groups of reducing sugars and the amine groups present in proteins, lipids, and nucleic acids. It ultimately results in the formation of advanced glycation end products (AGEs) [208]. It alters the molecular organization and charge distribution within the fibrils, which, as demonstrated, has noteworthy mechanical and biological implications [209]. In collagen with high levels of glycation exhibits reduced cellular adhesion and affinity towards proteoglycans leading to increased stiffness [210,211]. This is primarily attributed to a reduction in accessible integrin-binding sites [212]. Furthermore, it triggers the Receptor for AGEs (RAGE) on cells, a factor associated with vascular diseases and other complications related to diabetes [213].

There are 29 different types in the collagen superfamily discovered thus far. Collage I being the most abundant is present in the skin [214]. These different collagen types contain at least one triple helix domain and form a diverse range of supramolecular assemblies in the ECM. Several α chains and molecular isoforms and the use of alternative promoters and alternative splicing in the collagen biosynthesis further increase the diversity in the superfamily [215]. The diverse functionality of collagen is depicted by the existence of different collagen types and the diverse supramolecular structures they form. These suprastructures are biological composites that can have unique tissue-specific functional properties distinct from those of the macromolecules. These include fibrils, fibril associated collagens with interrupted triple helices (FACIT), FACIT-like collagens, and networks. Fibrils forming collagens include types I, II, III, V, XI, XXIV, and XXVII [216]. They form a major part of cartilage (collagens II, XI, and IX or of collagens II and III) [217], skin (collagens I and III) and cornea (collagen I and V) [218]. The fibrillar structure provides structural support to the tissues and bear large tensile loads [210]. Interestingly, varying tensile load is associated with the change in collagen fibril architecture in the skin. Skin-bearing high tensile loads have larger fibril diameters compared to the low loads [219,220]. Furthermore, the arrangement of collagen fibrils controls the exposure of integrin-recognizing motifs (GXOGER) sequence that affects integrin-mediated cellular adhesion and cell signaling [221]. FACITs include collagen IX, XII, XIV, and XX [222]. This type of suprastructure serves as the molecular bridge essential in the structurization of ECM whose peculiarities determine the inter-tissue differences in regulating the sizes of banded collagen fibrils [223]. FACIT-like collagen structures have commonality with FACIT but are structurally and functionally different from them. This group includes collagen XVI, XIX, XXI, and XXII [224]. These are located at the interfacial and basement membrane zones and differentiate types of tissues. These can bind to the fibrillin-1 at the epidermal–dermal basement membrane interface in the skin [225]. Lastly, collagen IV forms the networks in the basement membrane, resembling a chicken-wire-like structure. Diverse basement membranes exist depending on the anatomical site and circumstance-based differential expression. Collagen VI forms beaded filaments, broad-banded structures, and hexagonal networks [222]. It interacts with a diverse range of extracellular molecules, including collagens I, II, IV, and XIV, HA, heparin, and fibronectin, as well as integrins and the cell-surface proteoglycan NG2. Collagen VI structures play a role in maintaining tissue homeostasis, wound healing, and repair processes by influencing cell migration, differentiation, proliferation, and apoptosis [226]. Moreover, basement membrane having collagen IV, laminin, and fibronectin [227] allows endothelial and epithelial cell adherence, integrin-mediated migration and mechanotransduction through stiffness sensing [228].

Collagen and its metabolic derivatives can attract fibroblasts to the wound, facilitate migration, and enhance skin repair [229]. Collagen also promotes the synthesis of other types of collagens within the wound bed [230], fibroblast proliferation, and angiogenesis [231]. For these reasons, collagen-based materials such as sponges, injectables, hydrogels, skin grafts, wound dressings, and mimicking peptides are developed for wound healing and tissue engineering applications [232]. When used as a scaffold, collagen has physico-chemical properties that influence wound healing outcomes, as seen in a study where increased fiber dispersion reduced and led to more uniform wound contraction and higher cell infiltration in the wound bed. Therefore, collagen fiber stiffness and density are essential for controlling contraction while designing wound healing scaffolds [233]. Collagen-based materials are also successful controlled-release platforms for bioactives to enhance healing in diabetic wounds. A collagen-based injectable hydrogel loaded with umbilical cord stem cell factor (SCF) was used for diabetic tissue regeneration. Both hydrogel groups, with and without SCF, exhibited a high collagen deposition and neovascularization in a whole cortex defect model of a diabetic rat. In addition, the shift of macrophage phenotype towards M2 like and the reduction in TNF-α also existed in both hydrogel groups [234]. Atelocollagen is a low-immunogenic derivative of collagen obtained by enzymatic removal of the telopeptides from collagen I. Atelocollagen hydrogels sensitive to protease have been found to reduce MMP-9 activity by approximately 50% and promote wound closure by up to 99% by accelerating neo-dermal tissue formation [235]. In addition to collagen I being used for its wide range of applications in diabetic wound healing, collagen III has also been utilized recently due to its bioactivity. Collagen III is essential in cellular migration and proliferation, as seen during fetal scarless healing [236]. Not only this, the cell proliferative and migrative ability of fibroblasts and endothelial cells during the remodeling phase was demonstrated where recombinant collagen III was spatiotemporally and sequentially released from a composite hydrogel in an infected in vivo chronic wound model [237]. Moreover, as a composite biomaterial platform, the recent development of nanocomposite injectable collagen/chitosan hydrogel incorporated with MMP-9 inhibitor (ND-336)-loaded phycocyanin nanoparticles was reported as having improved diabetic wound healing effects through enhanced cell migration and reduced MMP-9 expression [199]. Overall, as an inherently bioactive biomaterial, collagen can promote cell proliferation, act as a biodegradable substrate for cell migration and ECM deposition, and modulate proteases and immune response [238].

The intricate bioarchitectural characteristics of collagen, as the primary constituent of the ECM, wield profound implications for its role in diabetic wound management (Figure 5). Variations in molecular composition, crosslinking, and suprastructural organization influence its ability to resist mechanical stress and proteolytic degradation and modulate cellular adhesion, migration, and proliferation within the wound microenvironment. Furthermore, chemical modifications such as glycation impact its biomechanical properties and cell signaling pathways. The diverse array of collagen types and supramolecular assemblies further accentuates its functional versatility, enabling tissue-specific adaptations and responses. Leveraging this interplay between collagen’s structure and function, novel biomaterials and scaffolds can be engineered to promote cell proliferation, ECM deposition, and tissue regeneration, ultimately fostering accelerated wound closure and improved clinical outcomes.

## 3. Conclusions and Future Outlook

The relationship between the structure and function of bioactive polymeric materials is not merely a theoretical concept; it is a well-established framework. This framework has been thoroughly explored in the context of tissue repair and regeneration, focusing on biopolymers such as alginate, chitosan, HA, and collagen in this article. These biopolymers are commonly employed in their pure forms or as composite materials for the treatment of diabetic wounds, as described here. Depending on the specific biopolymer in question, the structural characteristics considered can vary, encompassing factors like MW, DP, and modifications to the chemical structure for polysaccharides (alginate, chitosan, and HA). Focusing on proteins such as collagen extends to more complex, higher-order hierarchical structures that can influence their bioactivity. Consequently, understanding the intrinsic link between the structure and function of bioactive biopolymers is essential for comprehending their roles in biological processes. The limitation lies in the incomplete comprehension of the mechanisms that govern this structure–function relationship. It is worth noting that in many cases, the physico-chemical properties of these biopolymers, whether obtained through synthesis, modification, or procurement from suppliers, often require more extensive characterization. Moreover, information regarding the raw biopolymers purchased from vendors may be limited. The application of an integrated design approach that combines experiments and computational modeling for biopolymers is uncommon due to challenges in correlating insights obtained across varying length and time scales [239]. Consequently, recognizing the profound relationship between structure and function in synthesizing biopolymers is critical during the developmental stage.

By leveraging the structural dependence of bioactivity, it becomes possible to finely adjust physical and chemical parameters to direct the synthesis of biomaterial systems. However, it is essential to acknowledge that the mechanism governing the structure–function relationship of these biopolymers is an ongoing subject of exploration, and further research is imperative to deepen our understanding in this area.

Recent advancements in synthesizing composites involving various bioactive biopolymers have created materials with diverse properties tailored for specific wound healing requirements. Considering the factors of the diabetic wound microenvironment, a biomaterial composite system tailored explicitly for chronic wounds has been developed. For instance, diabetic wound healing has been successfully addressed using a range of combinations, such as HA/alginate membranes [240], HA/carboxy methyl chitosan gels [241], chitosan–alginate hydrogel [242,243], collagen–chitosan gels [133], collagen–HA hydrogels [244], and more. Additionally, the emergence of functional and intelligent biomaterials offers the potential for guided therapies that can lead to highly efficient outcomes in the healing of chronic wounds. It is important to note that using the same biopolymer in different structural forms can have advantages and disadvantages within the tissue healing timeline. Combining different forms of the same biopolymer in strategic locations can create advanced bioactive materials that align with desired healing outcomes over time. Biomaterials can be converted into different forms, such as nanoparticles, hydrogels, and scaffolds, or even engineered as a combination of these forms. The physical properties of these materials, such as their porosity, stiffness, viscosity, and shape, can significantly affect their biological functionality, which is essential to consider when using them for clinical applications.

## Figures and Tables

**Figure 1 biomimetics-09-00275-f001:**
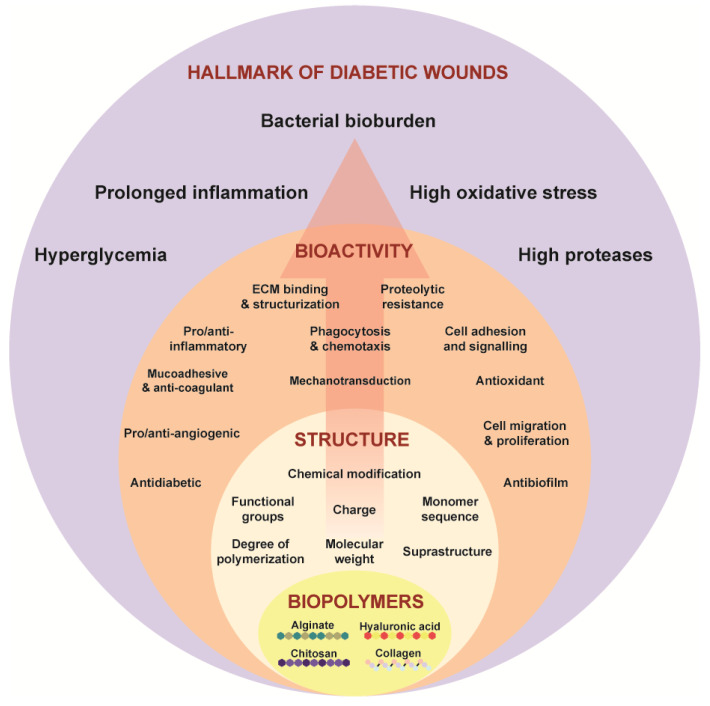
Structure–function paradigm as shown in Equation (1) in the biopolymers—alginate, chitosan, hyaluronic acid, collagen targeting the hallmarks of chronic wounds.

**Figure 2 biomimetics-09-00275-f002:**
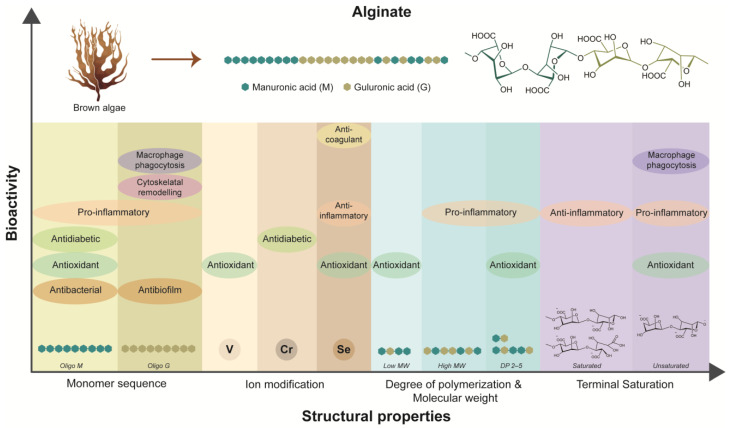
Dependency of inherent bioactivities of alginate on its structural properties.

**Figure 3 biomimetics-09-00275-f003:**
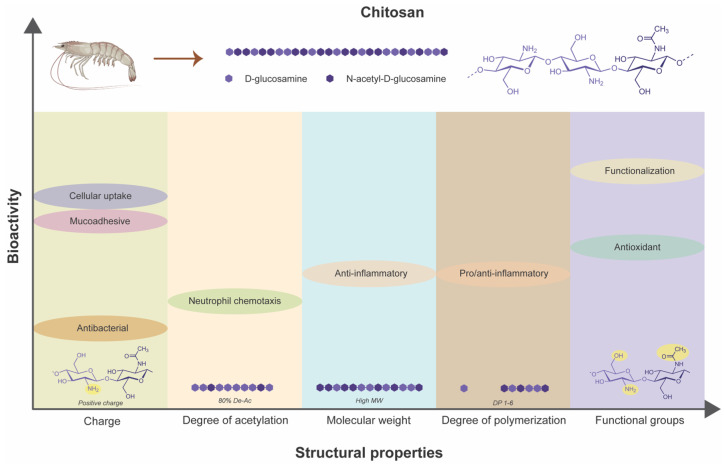
Dependency of inherent bioactivities of chitosan on its structural properties.

**Figure 4 biomimetics-09-00275-f004:**
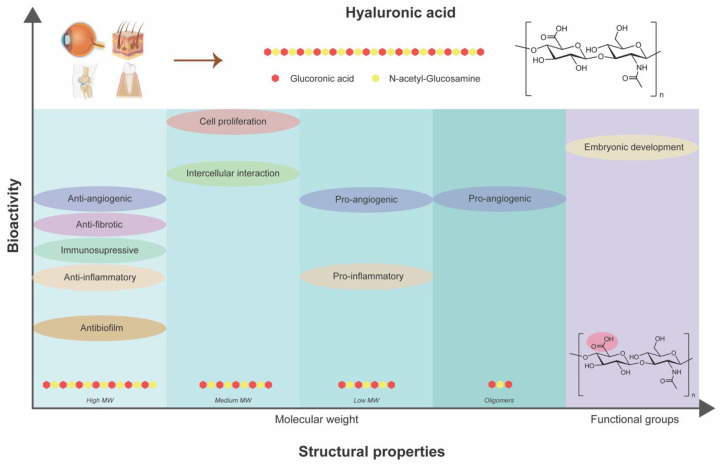
Dependency of inherent bioactivities of hyaluronic acid on its structural properties.

**Figure 5 biomimetics-09-00275-f005:**
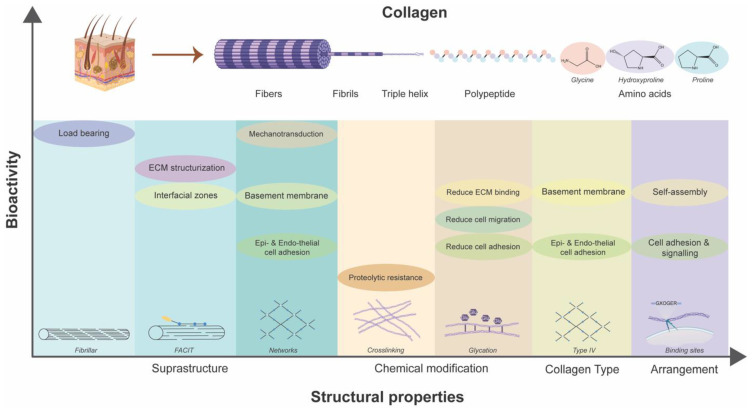
Dependency of inherent bioactivities of collagen on its structural properties.

## Data Availability

No new data were created or analyzed in this study. Data sharing is not applicable to this article.

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
