# Peer review of "Bioarchitectural Design of Bioactive Biopolymers: Structure–Function Paradigm for Diabetic Wound Healing"

_biomimetics, 2024, doi:10.3390/biomimetics9050275_

Round 1
Reviewer 1 Report
Comments and Suggestions for Authors
The manuscript entitled "BioArchitecture design principles for bioactive biopolymers: Harnessing the structure-function paradigm for diabetic wound healing" has been well-organized and well-defined. However some minor concerns should be addressed before acceptance.
1. Section 2 .1, 2.2, 2.3 & 2.4 require conclusion for consequence of bioploymers in chronic wounds.
2. for chitosan and alginate, the abbreviation should be provided for better presentation.
3. nanofiber-based wound dressings for chronic wounds should be provided for main sections.
4. limitation should be provided for section 3.
Comments on the Quality of English LanguageMinor editing of English language required
Reviewer 2 Report
Comments and Suggestions for Authors
Generally, the review is well structured and the informations are of interest for researches in this field, but some observations are necessary:
Line 125 – „Structure – function paradigme as shown in equation 2...”- In Figure 1 equation 2 is mentioned, but it is not written anywhere in the text. It may be about equation 1?
In the text, reference numbers should be placed in square brackets (instructions for authors), but a few situation require correction. For example:
Line 131 - [26], [28] correction [26, 28]
Line 133 - [29], [30] correction [29, 30].
Line 149, 159,
Line 188 [40], [38] correction [38, 40] and so on. Please, revise throughout the text.
Comments on the Quality of English LanguageEnglish revision is needed.
Reviewer 3 Report
Comments and Suggestions for Authors
See attached file

Comments on the Quality of English LanguageEnglish used is correct and readable
Round 2
Reviewer 3 Report
Comments and Suggestions for Authors
The authors responded to all my comments and improved the manuscript, which can now be recommended for publication.